# Generate What Matters: Steering Diffusion Models for Targeted Data Generation to Improve Classification

## Abstract

When labeled data are scarce, augmenting training sets with images from off-the-shelf generative models can help, but simply producing more samples is often insufficient. A major limitation of existing approaches is that they overlook the usefulness of synthetic data for a given classification task, evaluating generations only retrospectively through downstream performance. To address this issue, we identify the properties that make samples effective for classification and propose a principled way to generate them. We quantify a sample's usefulness through its *influence*, captured by how the classifier's loss gradient on that sample aligns with gradients from validation examples. Our key finding is that effective samples exhibit a clear **Class-Contrastive Influence** (C2I) gap: their gradients show strong positive alignment with same-class data and strong negative alignment with other-classes data. Our theoretical analysis confirms such high-gap samples are typically *hard examples* located near the decision boundary, which are valuable for improving model robustness. Building on this insight, we introduce a reinforcement-learning fine-tuning scheme for diffusion models with a C2I-based reward that drives generation of class-informative, boundary-proximal samples. Across several few-shot medical imaging tasks, C2I-guided generation consistently improves both accuracy and robustness over diffusion-based baselines, demonstrating that boundary-focused augmentation provides a principled and effective strategy in low-data regimes.

## 1 Introduction

Data augmentation is a long-standing tool for improving image classification when labeled data are scarce. Recent progress in diffusion models (Rombach et al., 2022) has renewed interest in learning with synthetic data by generating class-conditioned images to expand training sets. A growing body of work explores diffusion-based augmentation for classification by (i) perturbing real images and denoising them through a pretrained diffusion model to obtain diverse, label-preserving variants (Huang et al., 2024; Zhang et al., 2023; Wang et al., 2024a) and (ii) fine-tuning the generator on limited in-domain data to reduce distribution shift from the pretraining corpus (Kim et al., 2024; Zhang et al., 2023; Wang et al., 2024b). While these approaches often help, their gains are inconsistent, especially in few-shot and domain-shifted settings such as medical imaging, where class boundaries can be subtle and the target distribution diverges from internet-scale data used to train the generator.

We argue that a central limitation of current practice is the implicit assumption that more synthetic images with higher fidelity and diversity translate into better classifiers. In reality, not all generated examples are equally valuable for training. As depicted in Figure 1, the addition of certain subsets of generated images can substantially improve classification accuracy, whereas others offer little to no benefit despite appearing visually similar. Current pipelines rarely assess

| Training Datasets | Validation AUC |
|---|---|
| Original Only | 0.827 |
| Original + Set 1 | **0.853** |
| Original + Set 2 | 0.821 |

Figure 1: We fine-tune Stable Diffusion on 32 BreastMNIST images per class and generate 20 synthetic images per class for Sets 1 and 2. `Original only` uses only real images. Set 1 produces stronger classification AUC than Set 2.

Figure 2: **Overview of Class-Contrastive Influence (C2I).** Gradients from generated and validation samples are compared, and the C2I reward is computed from their alignment: encouraging positive alignment with same-class validation gradients and negative alignment with opposite-class gradients. This reward, defined jointly by the classifier, generated data, and validation data, guides reinforcement learning fine-tuning of the generator toward decision-boundary-proximal samples.

or optimize the task usefulness of synthetic data during generation; instead, usefulness is judged only retrospectively through downstream validation. This disconnect is particularly consequential in the few-shot regime, where every added image should carry a maximal learning signal.

To address this, we first characterize what makes a synthetic sample effective for classification.

**A criterion for sample effectiveness.** We propose evaluating the usefulness of synthetic samples not by generic generative metrics (e.g., FID, diversity) but by their *influence* on the downstream classifier. Concretely, we quantify usefulness via *influence score* (Koh & Liang, 2017): the alignment between the classifier's loss gradient on the generated sample and the gradients on validation examples. We find effective synthetic examples exhibit a pronounced *Class-Contrastive Influence* (C2I) yielding strong positive influence score with same-class validation examples and distinctly negative influence score with other-class examples. Our theoretical analysis supports this phenomenon, showing that *maximizing C2I draws features toward the validation set global mean, which lies close to decision boundary.* This property correlates with sample hardness, and training with such high-gap samples encourages the classifier to refine its decision boundary and achieve stronger generalization.

**RL fine-tuning diffusion models for effective data augmentation.** Building on this insight, we propose a general fine-tuning scheme that turns an off-the-shelf diffusion model into a targeted data generator for classification, as depicted in Figure 2. We design a reward based on C2I that scores generations by the degree to which their induced classifier gradients align with same-class validation gradients and oppose other-class gradients. We then fine-tune the generator with reinforcement learning (RL) to maximize C2I, steering toward hard, class-informative regions of the data manifold. The method is plug-and-play, requiring no architectural modifications to the generator or the classifier, and uses only a validation split to compute influence signals. Unlike prior approaches that focus on fidelity or diversity, C2I explicitly optimizes for task usefulness, enabling the generator to produce examples that sharpen decision boundaries.

**Scope and setting.** We evaluate this targeted augmentation in few-shot medical image classification, a regime where label scarcity and distribution shift routinely limit performance and where realistic yet task-useful synthetic data can be especially valuable. We compare against strong baselines, including standard transformation-based and other diffusion-based augmentation methods. Across multiple datasets with limited labels, our C2I-guided generator consistently improves accuracy and robustness, yielding models that generalize more effectively under domain shift.

**Contributions.**

- We formalize effectiveness through *Class-Contrastive Influence (C2I)* and show that high-C2I samples correspond to hard examples.

- We propose an RL-based scheme that leverages C2I as a reward to optimize diffusion models for generating boundary-proximal, class-informative samples.

- On multiple few-shot tasks, C2I consistently outperforms standard and diffusion-based augmentation baselines, establishing it as a principled strategy for low-data regimes.

Together, our study suggests a shift in using diffusion models for augmentation: rather than relying on fidelity and diversity, we should optimize generation for its downstream influence on the classifier. By aligning synthetic data with validation gradients, C2I turns off-the-shelf diffusion models into targeted generators that strengthen decision boundaries where labeled data are most scarce.

## 2 PRELIMINARY

This section outlines the key concepts underlying our method. Section 2.1 defines notation, Section 2.2 reviews gradient-based influence estimation, and Section 2.3 presents the RL framework for fine-tuning diffusion models.

### 2.1 NOTATION

Let the number of classes be $K \geq 2$. We write the training set as $\mathcal{D} = \bigcup_{c=1}^{K} \mathcal{D}_c$ and the validation set as $\mathcal{V} = \bigcup_{c=1}^{K} \mathcal{V}_c$, where $\mathcal{D}_c$ and $\mathcal{V}_c$ contain samples of class $c$. We use small Latin letters for vectors (e.g., image samples and feature vectors) $x, v, h \in \mathbb{R}^d$, and Greek letters $\alpha, \beta, \ldots$ for scalars. The reward is computed over *sets* of images: a set is a batch $\mathbf{x} = \{x^{(j)}\}_{j=1}^{n}$ generated by the diffusion model, conditioned on a specific class $c$; thus $\mathbf{x} \subset \mathcal{D}_c$. Validation samples are denoted by $v \in \mathcal{V}_c$. We denote the diffusion model by $\epsilon_\theta$ and the ViT-based classifier (Dosovitskiy et al., 2020) by $f_\phi$. The cross-entropy loss with respect to $f_\phi$ is written as $\ell(x; \phi)$. The cross-entropy loss for $K$-way classification is written as $\ell(x; \phi)$.

### 2.2 GRADIENT-BASED INFLUENCE ESTIMATION

Pruthi et al. (2020) quantifies a training example's influence by tracking its impact on validation loss using gradient information. If $x$ is a training sample, $v$ a validation sample, the change in the validation loss by a single parameter update can be approximated as

$$\ell(v; f_{t+1}) - \ell(v; f_t) = -\eta_t \langle \nabla\ell(x; f_t), \nabla\ell(v; f_t) \rangle + \mathcal{O}(\|\nabla\ell(v, f_t)\|^2), \qquad (1)$$

where $f_t$ represents the model at time step $t$, and $\ell(\cdot; \cdot)$ is the loss function. This approximation suggests that if the loss gradients are positively aligned (i.e. $\nabla\ell(x) \sim \gamma\nabla\ell(v), \gamma > 0$), the training sample is maximally effective in reducing validation loss.

Xia et al. (2024) extend this observation to Large Language Models (LLMs) with several modifications: (1) adapting gradient estimation to the Adam optimizer, (2) normalizing with cosine similarity, (3) computing only LoRA Hu et al. (2022) gradients for efficiency, and (4) applying random projection Park et al. (2023) for dimensionality reduction. We adopt these modifications to compute influence, as they are well suited for large transformer-based models.

Specifically, for a classification model $f_\phi$, let $\tilde{\nabla}\ell \in \mathbb{R}^Q$ and $\tilde{\Gamma} \in \mathbb{R}^Q$ denote the SGD and Adam optimizer LoRA gradients, respectively. The choice of optimizer (SGD or Adam) is not fundamental to our method. We used the Adam optimizer in our experiments, as it is commonly employed for training ViT classifiers.

Given a set $\mathbf{x} \in \mathcal{D}_c$ of training samples and a validation sample $v \in \mathcal{V}_{\bar{c}}$ for some $c, \bar{c} \in \{0, 1\}$, we compute *Influence* as

$$\mathcal{A}^\phi(\mathbf{x}, v) \triangleq \cos\left(\nabla\ell(v; \phi), \Gamma(\mathbf{x}; \phi)\right), \qquad (2)$$

where LoRA gradients are projected into a low $q$-dimensional space via random projection $\Pi \in \mathbb{R}^{Q \times q}$, such that $\nabla\ell(v; \phi) = \Pi^\top \tilde{\nabla}\ell(v; \phi)$.

### 2.3 REINFORCEMENT LEARNING FRAMEWORK FOR FINE-TUNING DIFFUSION MODELS

Reinforcement learning (RL) fine-tuning enhances diffusion models by optimizing generation through reward feedback rather than likelihood maximization. The objective is to maximize the expected reward, i.e. $\mathcal{L}_{\text{RL}} = \mathbb{E}_{p_\theta(x)}[r(x)]$, optimized using denoising diffusion policy optimization (DDPO) (Black et al., 2023). To support multi-step updates, DDPO uses importance sampling,

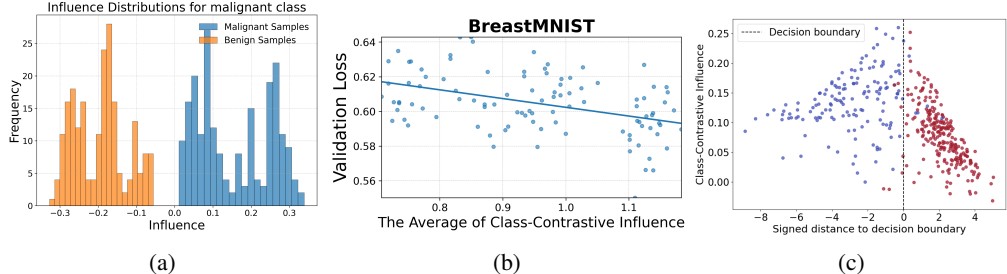

(a)                                       (b)                                       (c)

Figure 3: (a) With the BrestMNIST dataset, we compute *influence* between synthesized *malignant* images and validation samples. They show positive scores with *malignant* and negative with *benign*, demonstrating clear separation. (b) The average of Class-Contrastive Influence (C2I) is negatively correlated with validation loss ($r = -0.241$, $p = 0.015$) (c) Relationship between C2I and the signed distance to the decision boundary in logistic regression. Samples with higher C2I tend to lie closer to decision boundary, corresponding to harder examples.

resulting in the gradient:

$$\nabla_\theta J_{\text{DDRL}} = \mathbb{E}\left[\sum_{t=1}^{T} \frac{p_\theta(x_{t-1}|x_t, C)}{p_{\theta_{\text{old}}}(x_{t-1}|x_t, C)} \cdot \nabla_\theta \log p_\theta(x_{t-1}|x_t, C)\, r(x_0, C)\right],$$  (3)

where $\theta$ is the model, $C$ the context, $x_t$ the intermediate state, and $x_0$ the final output[1]. This formulation aligns diffusion models with task-specific objectives, enabling preference-guided generation.

## 3 THE PROPOSED METHOD

In this section, we introduce a principled data augmentation strategy to improve classification performance. We argue that the usefulness of a synthetic sample should be judged by its *influence* on the downstream classifier, and that this criterion should guide the generation process to enable targeted data generation.

To this end, we formalize what makes training samples effective through the notion of *influence* and propose an RL-based fine-tuning framework for diffusion models that encourages the generation of such samples. Section 3.1 defines *class-contrastive influence* as a key property of effective samples, Section 3.2 establishes its theoretical connection to sample *hardness*, and Section 3.3 presents the RL fine-tuning framework. An overview of our approach is provided in Figure 2.

### 3.1 CLASS-CONTRASTIVE INFLUENCE AS A KEY PROPERTY OF EFFECTIVE SAMPLES

We begin by analyzing what makes a sample effective for classification, using gradient-based influence (defined in equation 2) to quantify its impact on validation loss. While selecting data with high influence scores is a successful strategy in some settings like fine-tuning LLMs (Xia et al., 2024), this principle can fail in classification. The reason is that the validation set contains conflicting signals from different classes; a sample that helps one class may harm another. As a result, raw, class-agnostic influence scores show no correlation with classification performance, as illustrated in Appendix Figure 6.

To understand this failure, we analyze influence scores on a per-class basis. As shown in Figure 3a, we find a distinct, class-contrastive pattern: a sample's influence is consistently positive on validation data from its own class and negative on data from other classes. Theoretical analysis in Appendix A.1 confirms that this phenomenon holds more generally. This insight leads us to hypothesize that a sample's value lies not in its overall influence, but in its ability to create a large separation, or gap, between the influence distributions of different classes.

To formalize this, we propose quantifying this separation in a binary classification task. For each synthetic sample $\mathbf{x}$ (conditioned on one class) and validation set $\mathcal{V}_c$ from class $c \in \{0, 1\}$, we

---

[1]With a slight abuse of notation, we find it convenient to use the same notation $x_0$ to denote the class membership $x \in \mathcal{D}_0$ in later sections.

compute the class-specific mean and variance of influence:

$$\mu_c(\mathbf{x}, \mathcal{V}_c) \triangleq \frac{1}{N} \sum_{j=1}^{N} \mathcal{A}^\phi(\mathbf{x}, v_c^j), \quad \sigma_c^2(\mathbf{x}, \mathcal{V}_c) \triangleq \frac{1}{N-1} \sum_{j=1}^{N} \left( \mathcal{A}^\phi(\mathbf{x}, v_c^j) - \mu_c \right)^2, \quad c = 0, 1, \quad (4)$$

in which $\mathcal{A}^\phi$ is defined in equation 2.

Using these statistics, we introduce the *Class-Contrastive Influence* (C2I) score, which measures the influence gap:

$$C2I(\mathbf{x}, \mathcal{V}; \phi) = \frac{(\mu_0 - \mu_1)^2}{\sqrt{\sigma_0^2 + \sigma_1^2}}, \tag{5}$$

which penalizes distributional overlap while amplifying mean separation, making it analogous to a class-separability score. **This formulation naturally extends to multi-class classification; see Appendix A.3.**

Crucially, experimental results in Figure 3b confirm a consistent correlation between the C2I score and classification effectiveness (See Appendix B.1 for experimental details). This establishes that *the influence gap between classes is a key determinant of sample usefulness.* This naturally raises the next question: *why are samples with a large influence gap particularly effective?*

In the next section, we uncover the property of C2I that explains their value for training.

### 3.2 UNDERSTANDING CLASS-CONTRASTIVE INFLUENCE THROUGH SAMPLE HARDNESS

In this section, we theoretically and empirically show that samples with large *Class-Contrastive Influence (C2I)* improve classification performance due to their connection to *sample hardness*.

**Theoretical evidence.** We first show that, in the context of logistic regression, maximizing $C2I$ induces a feature-averaging effect that moves samples closer to the decision boundary—a known property of hard examples (Srinidhi & Martel, 2021).

**Theorem 1.** *Consider a Logistic Regression model with output probabilities $p(x) = Sigmoid(w^\top x)$, $x, w \in \mathbb{R}^d$ and cross-entropy loss $\ell(x)$. Let $\mathcal{V}_0$ and $V_1$ be validation sets containing $N/2$ samples from class 0 and class 1 respectively and $\mathcal{V} = \mathcal{V}_0 \cup \mathcal{V}_1$ (for even $N$). Consider the cosine similarity between the loss gradients of a sample $x$ and a validation sample $v_i$ as:*

$$\mathcal{A}(x, v_i) = \frac{\nabla \ell(x) \cdot \nabla \ell(v_i)}{\|\nabla \ell(x)\| \, \|\nabla \ell(v_i)\|},$$

*and define $\mu_c(x) = \frac{1}{N} \sum_{i \in \mathcal{V}_c} \mathcal{A}(x, v_i)$. Then the sample $x$ that maximizes the absolute class influence gap $|\mu_0(x) - \mu_1(x)|$ is given by the convex combination*

$$x^\star = \sum_{v_i \in \mathcal{V}} \alpha_i v_i, \qquad \alpha_i = \frac{1/\|v_i\|}{\sum_{v_j \in \mathcal{V}} 1/\|v_j\|}. \tag{6}$$

Here $\alpha_i > 0$ for all $i = 1, \cdots, N$ and $\sum_i \alpha_i = 1$. When the variance of $|v_i|$ is small (e.g., when the origin lies far from the data clusters), $\alpha_i \approx 1/N$ and $x^\star$ approaches the global average of the validation set.

This averaging effect directly relates to hardness. In logistic regression, if the class means $\bar{v}_0$ and $\bar{v}_1$ are predicted correctly (i.e., $p(\bar{v}_0) > \frac{1}{2}$ and $p(\bar{v}_1) < \frac{1}{2}$), then the global average $\bar{v} = \frac{1}{2}(\bar{v}_0 + \bar{v}_1)$ exhibits higher loss and lower confidence than either class mean (see Lemma 2 in Appendix A). *Thus, Theorem 1 implies that maximizing $C2I$ drives features toward the global mean across classes in the validation dataset, naturally producing harder examples.*

**Empirical evidence.** Figure 3c empirically confirms this link: in a logistic regression setup, samples with higher $C2I$ lie closer to the decision boundary, supporting their interpretation as hard examples. In this toy experiment, logistic regression is trained on the Breast Cancer dataset (Wolberg et al., 1993), where the original 30-dimensional features are reduced with PCA for visualization. The distance to the decision boundary is measured by the classification logit. We further validate this observation in a larger-scale setting with our ViT classifier. As training progresses and $C2I$

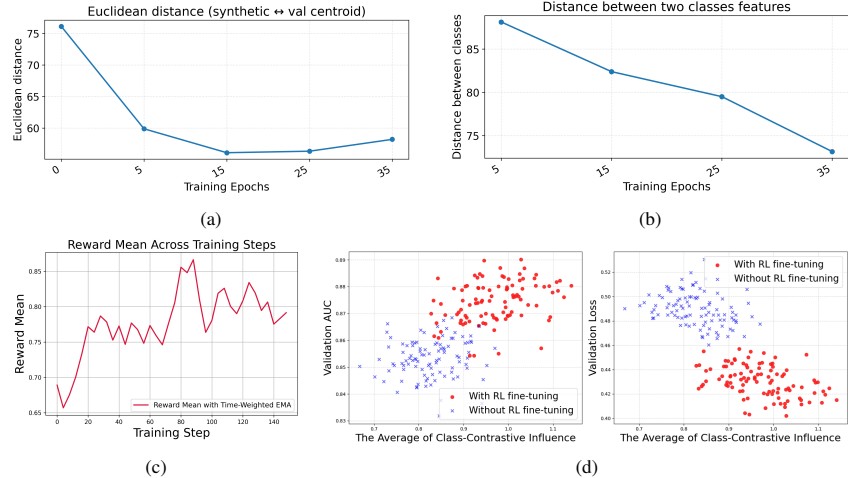

Figure 4: (a) During RL fine-tuning, increasing class-contrastive influence (C2I) pulls synthetic features closer to the mean of validation features. (b) The inter-class feature distance of synthetic images decreases during RL fine-tuning, indicating greater feature similarity. (c) The average C2I reward steadily increases during RL fine-tuning. (d) Samples generated after RL fine-tuning exhibit higher C2I, improved AUC, and lower validation loss.

increases, features of diffusion-generated synthetic images move closer to the centroids of validation features (Fig. 4a). At the same time, the distance between synthetic class clusters decreases (Fig. 4b), indicating that features from different classes become more aligned.

These theoretical and empirical findings establish that high $C2I$ is connected to sample hardness. Since prior work (Shrivastava et al., 2016; Hacohen & Weinshall, 2019; Song et al., 2024; Srinidhi & Martel, 2021; Liu et al., 2017; Yuan et al., 2022) has shown that hard examples improve robustness and generalization, our results explain the trend in Figure 3b: maximizing $C2I$ guides diffusion models to generate harder, useful samples that help classifiers refine their decision boundaries.

### 3.3 Maximizing Class-Contrastive Influence in Diffusion Models via RL

Motivated by the previous findings, we hypothesize that generating images with high C2I can improve classification performance. To this end, we fine-tune a pre-trained T2I diffusion model within a reinforcement learning (RL) framework that encourages the generation of such samples.

**Few-shot learning setup and model preparation.** We consider a few-shot learning scenario where the training set $\mathcal{D}$ contains only a few labeled samples per class. Both the diffusion model $\epsilon_\theta$ and the classifier $f_\phi$ are trained on $\mathcal{D}$, as detailed in Sec. 4. For the classifier, we save a checkpoint after $a$ epochs, denoted $\phi_a$, which is later used for gradient computation.

**RL fine-tuning procedure.** We precompute validation gradients for each class using the fine-tuned classification model $f_{\phi_a}$. Specifically, we collect the set of projected validation gradients for each class as $\mathbf{G}_c^{\text{val}} = \left\{ \nabla\ell(v; \phi_a) \middle| v \in \mathcal{V}_c \right\}, c = 0, 1$. During RL fine-tuning, we compute projected gradients of generated samples $\Gamma(x; \phi_a)$ on-the-fly. Then, $\Gamma(\mathbf{x}; \phi_a), \mathbf{G}_c^{\text{val}}$, are used to compute the reward based on C2I. The reward for the $i$-th set of generated samples $\mathbf{x}^i$ is then defined as:

$$r(\mathbf{x}^i, \mathcal{V}; \phi_a) = C2I(\mathbf{x}^i, \mathcal{V}; \phi_a) = \frac{\left(\mu_0(\mathbf{x}^i, \mathcal{V}_0) - \mu(\mathbf{x}^i, \mathcal{V}_1)\right)^2}{\sqrt{\sigma^2(\mathbf{x}^i, \mathcal{V}_0) + \sigma^2(\mathbf{x}^i, \mathcal{V}_1)}}. \tag{7}$$

In practice, the same reward is assigned to all samples in each generated $\mathbf{x}^i$. Each minibatch during fine-tuning consists of multiple such sets: $\mathbf{x}_{\text{minibatch}} = \left\{ \mathbf{x}_c^1, \ldots, \mathbf{x}_c^n, \mathbf{x}_{\bar{c}}^1, \ldots, \mathbf{x}_{\bar{c}}^n \right\}$. The overall RL objective is defined as the expected reward across the distribution of generated samples:

$$\mathcal{L}_{\text{RL}} = \mathbb{E}_{p_\theta(x)} \left[ r(x, \mathcal{V}; \phi_a) \right]. \tag{8}$$

Table 1: AUC scores across datasets and augmentation methods. + denotes the training data augmentation. P.MNIST denotes PneumoniaMNIST. Bold values indicate the best performance in average.

| Backbone | Method | BreastMNIST | DermaMNIST-binary | P.MNIST | Avg. |
|----------|--------|-------------|-------------------|---------|------|
| ViT | Original only | 0.828 | 0.846 | 0.941 | 0.873 |
| | + RandAugment | 0.858 | 0.824 | 0.954 | 0.879 |
| | + RandomErasing | 0.873 | 0.839 | 0.945 | 0.885 |
| | + Mixup | 0.823 | 0.845 | 0.890 | 0.867 |
| | + DataDream | 0.822 | 0.819 | **0.958** | 0.866 |
| | + Dataset Expansion | 0.844 | 0.852 | 0.943 | 0.880 |
| | + DistDiff | 0.764 | 0.805 | 0.938 | 0.784 |
| | + Ours | **0.885** | **0.853** | 0.945 | **0.894** |
| ResNet18 | Original only | 0.815 | 0.777 | 0.935 | 0.842 |
| | + RandAugment | 0.764 | 0.787 | 0.936 | 0.829 |
| | + RandomErasing | 0.758 | 0.747 | 0.900 | 0.802 |
| | + DataDream | 0.844 | 0.804 | 0.947 | 0.865 |
| | + Dataset Expansion | 0.804 | 0.831 | **0.956** | 0.864 |
| | + Ours | **0.854** | **0.836** | 0.956 | **0.882** |

## 4 EXPERIMENTS

In this section, we evaluate our method on few-shot medical image classification tasks.

**Few-shot setup and model preparation.** We adopt a few-shot regime with 16 or 32 labeled samples per class for training ($\mathcal{D}$) and use a validation set ($\mathcal{V}$) solely to provide gradient feedback for RL. The classifier $f_\phi$ is a ViT-B/16 pre-trained on ImageNet (Ridnik et al., 2021; Deng et al., 2009), and the generator $\epsilon_\theta$ is Stable Diffusion 2.1 (SD) (Rombach et al., 2022); both models are adapted to $\mathcal{D}$ using LoRA (Hu et al., 2022). For the diffusion model, we follow the fine-tuning protocol of Kim et al. (2024), updating LoRA weights on the linear projections within attention layers of both the text encoder and the U-Net.

**RL-guided diffusion fine-tuning.** We perform RL fine-tuning of SD (Black et al., 2023) guided by the C2I reward (equation 7). At each RL step, a ViT-B/16 trained on the few-shot set supplies the gradients used to compute the reward. We select the diffusion checkpoint with the highest average reward within the first 30 epochs and use it to synthesize 500 images per class for augmentation, following Kim et al. (2024). Examples of synthetic images generated by our method are provided in Appendix B.3.

**Classifier training.** To evaluate downstream performance, we train classifiers on datasets augmented by different methods. We assess cross-architecture generalization by training both ViT-B/16 (the backbone used during RL) and ResNet-18 (not used during RL), thereby testing whether RL-guided augmentation transfers to unseen model families. Each model uses standard optimization settings, detailed in Appendix B.3, and the best checkpoint is selected by validation AUC.

**Datasets and Evaluation.** We evaluate our approach on three MedMNIST benchmarks Yang et al. (2023): BreastMNIST, DermaMNIST (binary: DermaMNIST-binary; multi-class: DermaMNIST-all; see Appendix B.3 for details), and PneumoniaMNIST. To simulate a few-shot setting, we randomly sample 16 labeled examples per class for training (32 per class for BreastMNIST). Details of the validation set are provided in Appendix B.3. We evaluate the classification model under two test settings: (1) using the original clean test images, and (2) using noisy test images. The latter assesses the robustness of the learned decision boundary. We apply three types of input noise to the test images: salt-and-pepper noise (amount 0.01), Gaussian blur (radius 2), and JPEG compression (quality 25%). Given the class imbalance across all benchmarks, we report AUC as the primary evaluation metric, as it provides a more robust measure than accuracy.

**Baselines.** We compare against diffusion-based augmentation baselines including DataDream (Kim et al., 2024), Dataset Expansion (Zhang et al., 2023), and DistDiff (Zhu et al., 2024) as well as standard augmentation methods including RandAugment (Cubuk et al., 2020), RandomErasing (Zhong et al., 2020), and Mixup (Zhang et al., 2017). In addition, we evaluate a simple baseline for generating hard examples, where SD is fine-tuned solely on validation images that were misclassified by the classifier. Details of the experimental setup and results are provided in Appendix B.3.

Table 2: Robustness results (AUC) under different types of noise across three datasets. Bold values indicate the best performance in average.

| Dataset | Noise Type | Original only | Dataset Exp. | DataDream | Ours |
|---|---|---|---|---|---|
| DermaMNIST-binary | Salt&Pepper | 0.766 | 0.813 | 0.810 | 0.830 |
| | JPEG | 0.806 | 0.800 | 0.821 | 0.831 |
| | Blur | 0.827 | 0.848 | 0.828 | 0.841 |
| | Avg. | 0.800 | 0.820 | 0.820 | **0.834** |
| BreastMNIST | Salt&Pepper | 0.764 | 0.772 | 0.817 | 0.832 |
| | JPEG | 0.760 | 0.804 | 0.814 | 0.816 |
| | Blur | 0.758 | 0.727 | 0.765 | 0.810 |
| | Avg. | 0.761 | 0.768 | 0.799 | **0.819** |
| PneumoniaMNIST | Salt&Pepper | 0.868 | 0.861 | 0.823 | 0.792 |
| | JPEG | 0.922 | 0.907 | 0.950 | 0.940 |
| | Blur | 0.930 | 0.881 | 0.956 | 0.946 |
| | Avg. | 0.907 | 0.883 | **0.910** | 0.893 |

## 4.1 RESULTS

**Effect of RL fine-tuning with $C2I$.** First, we examine how $C2I$ evolves under RL fine-tuning. Figure 4c shows a steady increase in the mean reward, indicating that the diffusion model progressively generates samples with higher $C2I$. As training proceeds, the features of diffusion-generated images move closer to the centroids of validation features (Fig. 4a), while the distance between class clusters decreases (Fig. 4b), suggesting the generation of harder examples near the decision boundary. Finally, we evaluate whether this augmentation reduces validation loss and improves AUC. Details of the experimental setup are provided in Appendix B.1. As shown in Figure 4d, both metrics improve, confirming that our method produces more effective samples.

**Improving classification performance and robustness with augmented training data.** Table 1 shows that our method consistently outperforms existing augmentation strategies across datasets and backbones, achieving the higher average AUC overall. Whereas several baselines occasionally underperform relative to using only the original images, our approach reliably improves performance. Notably, the gains with ResNet-18 indicate that the synthesized samples are broadly informative and transfer beyond the backbone used during RL fine-tuning.

In addition, we evaluate model robustness under different types of noise. As shown in Table 2, our method achieves notable improvements in AUC under noisy conditions in both DermaMNIST-binary and BreastMNIST, and showing comparable results with DataDream in PneumoniaMNIST. These results suggest that the generated samples help establish a more stable and generalizable decision boundary. While Dataset Expansion and DataDream offer moderate gains, their performance is less consistent across noise types and datasets.

**Effect of RL training epochs on performance.** Table 3 reports the effect of RL fine-tuning duration on classification performance in BreastMNIST. Test AUC consistently increases with the number of RL training epochs, reaching 0.89 at epoch 20. This trend indicates that longer RL fine-tuning enables the diffusion model to generate more informative samples, thereby improving downstream classification performance.

**The effect of the number of synthesized images for augmentation.** The number of synthetic images per class is treated as a hyperparameter. Figure 5 shows that our method (green) consistently outperforms baseline approaches across all training sizes, achieving higher AUC. Its performance steadily improves as more synthesized data are added, demonstrating the effectiveness of our generation strategy. In contrast, DataDream and Dataset Expansion exhibit the oppo-

Table 3: AUC on BreastMNIST with varying RL fine-tuning epochs.

| Method | Epoch 10 | Epoch 15 | Epoch 20 |
|---|---|---|---|
| AUC | 0.83 | 0.86 | 0.89 |

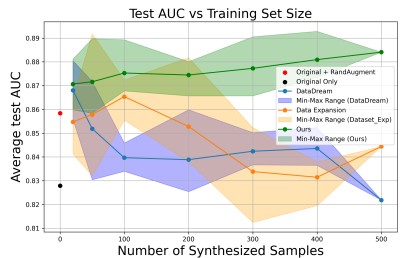

Figure 5: AUC with varying numbers (N) of synthesized images. Averaged over 5 random seeds.

site trend, adding more synthesized samples leads to a
decline in AUC.

**Generalization to multi-class classification.**   We evaluated our method on the DermaMNIST-all
dataset, which contains seven classes, using the multi-class formulation of C2I defined in equa-
tion 44 in Appendix A.3. As shown in Table 4, applying our approach leads to improved classifica-
tion accuracy.

Table 4: Test classification accuracy on the DermaMNIST-all dataset (7-class). Our method achieves
the highest accuracy compared to standard augmentation baselines and DataDream.

| Method | Original | Random Erasing | RandAugment | DataDream | Ours |
|---|---|---|---|---|---|
| **Accuracy** | 0.648 | 0.664 | 0.669 | 0.660 | **0.683** |

**Computation cost comparison.** For cost comparison, all experiments were conducted on a single
NVIDIA A100 GPU. RL fine-tuning ran for 30 epochs and required about 5 GPU hours, while clas-
sifier training on few-shot examples was lightweight, completing in roughly 10 minutes. Once RL
training was complete, image generation incurred no additional cost beyond standard SD sampling.
In contrast, Dataset Expansion introduced inference overhead, with each generated image requiring
an *additional*  25 seconds for its test-time optimization procedure.

## 5   RELATED WORK

**Diffusion models for data augmentation.** T2I diffusion models are widely used to improve clas-
sification by augmenting training data Du et al. (2023); Azizi et al. (2023); Islam et al. (2024); He
et al. (2022); Huang et al. (2024); Shipard et al. (2023); Wang et al. (2024a); Zhang et al. (2023);
Trabucco et al. (2023); Fu et al. (2024). A common strategy Zhang et al. (2023); Trabucco et al.
(2023); Fu et al. (2024); Islam et al. (2024); Wang et al. (2024a) is to add noise to original samples
and denoise them with pre-trained diffusion models, thereby enhancing diversity. However, these
approaches often target natural images close to the pretraining distribution (e.g., animals Kim et al.
(2024); Zhang et al. (2023); Wang et al. (2024b) or objects Krause et al. (2013)), limiting effective-
ness on out-of-distribution tasks. Others Kim et al. (2024); Zhang et al. (2023) fine-tune diffusion
models on small labeled sets to generate domain-aligned data. In contrast, we fine-tune T2I diffusion
models to explicitly improve the utility of generated samples for classification.

**Influence estimation from gradients.** Gradient-based influence estimation is widely used for data
selection Mirzasoleiman et al. (2020); Wang et al. (2020); Pruthi et al. (2020). We follow Pruthi
et al. (2020), who approximate training dynamics to estimate a sample's influence on held-out data.
Xia et al. (2024) recently applied this approach to select instruction-tuning data for LLMs, extending
it to Adam optimization and LoRA fine-tuning Hu et al. (2022). This method is also compatible with
ViT Dosovitskiy et al. (2020), which we adopt as our backbone.

**RL fine-tuning of diffusion models.** Reinforcement learning (RL) has been explored to fine-tune
diffusion models beyond supervised objectives. RL-based methods, such as RLHF, align generative
models with user preferences or domain-specific goals Black et al. (2023); Yang et al. (2024); Fan
et al. (2023).

## 6   CONCLUSION

We investigated how to fine-tune diffusion models to generate more effective training samples for
few-shot classification. Our analysis showed that the most useful samples exhibit a large influ-
ence gap between two classes: their gradients are aligned with validation samples from the same
class and misaligned with others. Leveraging this insight, we proposed a reinforcement learning
approach using a *Class-Contrastive Influence* reward. Our method effectively improve classification
performance across medical imaging tasks. However, our study has certain limitations. Our method
introduces additional computational overhead compared to using original training data alone.

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

## A ADDITIONAL DETAILS ON THE PROPOSED METHOD

### A.1 THEORETICAL EVIDENCE FOR THE OPPOSITE-SIGNED SIMILARITIES IN FIGURE 3A

In this section, we present theoretical evidence to explain the emergence of opposite-signed similarities across different class labels. Specifically, we explore the relationship between the gradient of the loss and the gradient of the feature vector in two different scenarios.

Consider a binary classification model in which the predicted classification probability of sample $x$ is given by $p(x) = \text{Sigmoid}(w^\top h(x; \theta))$. Here, $w \in \mathbb{R}^d$ is the classifier vector, $h(x; \theta) \in \mathbb{R}^d$ the feature vector corresponding to sample $x$ and depending on parameters $\theta$. We drop dependence on $\theta$ from the notation to reduce clutter.

First, we analyze gradients w.r.t the classifier head $w$ alone. For any two samples $x, x'$, we have the following relation between loss gradients.

$$\nabla_w \ell(x) \cdot \nabla_w \ell(x') = e(x)e(x')h(x) \cdot h(x'), \tag{9}$$

where

$$e(x) = \begin{cases} p(x) & x \in D_0 \\ p(x) - 1 & x \in D_1 \end{cases}. \tag{10}$$

This shows that if the two samples belong to opposite classes and their features have positive alignment, their loss gradients will be negatively aligned. In contrast, if the samples belong to the same class, their feature alignment has the same sign as their loss gradient alignment. If $h(x)$ is the output of a ReLU activation layer, as in the case of ResNet (cf. GeLU used in ViT), the features of any two samples will tend to be positively aligned, regardless of their class labels. As a result, the gradient alignment (w.r.t the classifier head) will agree with Figure 3a.

We now extend this analysis to gradients with respect to the parameters of the final layer in a ReLU-based feature extractor. Consider a feature extractor defined as:

$$h_i(x) = \text{ReLU}(\sum_j W_{ij} h_j^{(-1)}(x)), \tag{11}$$

and $h^{(-1)}$ is the representation in the next-to-last layer.

Gradient of the feature vector w.r.t $W$ takes the following form:

$$\nabla_W h_i(x) \equiv \frac{\partial h_i(x)}{\partial W_{jk}} = \delta_{ij} h_k^{(-1)}(x) \Theta(h_i(x)), \tag{12}$$

where $\Theta$ is the step function. For any two samples $x, x'$, the inner product between loss gradients will be

$$\nabla_W \ell(x) \cdot \nabla_W \ell(x') = e(x)e(x') \sum_{jk} \frac{\partial(w^\top h(x))}{\partial W_{jk}} \frac{\partial(w^\top h(x'))}{\partial W_{jk}}. \tag{13}$$

According to Eq. equation 12,

$$\frac{\partial(w^\top h(x))}{\partial W_{jk}} = w_j h_k^{(-1)}(x) \Theta(h_i(x)), \tag{14}$$

so that the dot product will be

$$\frac{\partial(w^\top h(x))}{\partial W_{jk}} \cdot \frac{\partial(w^\top h(x'))}{\partial W_{jk}} = (w_j)^2 h_k^{(-1)}(x) h_k^{(-1)}(x') \Theta(h_i(x)) \Theta(h_i(x')) \geq 0. \tag{15}$$

Using this result in Eq. equation 13, it is clear that $\nabla_W \ell(x) \cdot \nabla_W \ell(x')$ is positive for same-class samples and negative for opposite-class samples. This result further confirms the observation of opposite-signed similarities across different classes in Figure 3a.

### A.2 PROOF OF THEOREM 1

In this section, we provide proofs for Theorem 1 in Section 3.2 and exhibit more results.

**Lemma 1.** *In a Logistic Regression model with output probabilities $p(x) = Sigmoid(w^\top x)$, $x, w \in \mathbb{R}^d$ and cross-entropy loss $\ell(x)$, we have*

$$\cos\big(\nabla\ell(x), \nabla\ell(x')\big) = s(x)s(x')\,\cos(x, x'), \tag{16}$$

*with $s(x) = +1$ if $x$ is in class 0 and $s(x) = -1$ if in class 1.*

*Proof.* We begin by computing the gradient of the cross-entropy loss $\ell(x)$ for a sample $x$ with label $y \in \{0, 1\}$. Using $p(x) = \text{Sigmoid}(w^\top x)$, we have $\nabla_w \ell(x) = e(x)x$, where

$$e(x) = \begin{cases} p(x), & \text{if } x \in D_0, \\ p(x) - 1, & \text{if } x \in D_1. \end{cases} \tag{17}$$

The inner product of gradients between sample $x$ and a validation sample $v$ is:

$$\nabla\ell(x) \cdot \nabla\ell(v) = e(x)e(v)\, x \cdot v. \tag{18}$$

Define the class sign function as:

$$s(x) = \begin{cases} +1, & x \in D_0, \\ -1, & x \in D_1. \end{cases}$$

Assuming the gradients are nonzero, the cosine similarity is:

$$\cos\big(\nabla\ell(x), \nabla\ell(x')\big) = \frac{\nabla\ell(x) \cdot \nabla\ell(v)}{\|\nabla\ell(x)\|\|\nabla\ell(v)\|} = s(x)s(v)\frac{x \cdot v}{\|x\|\|v\|}. \tag{19}$$

$\square$

If $x$ and $x'$ belong to opposite classes but exhibit strong feature alignment, their loss gradients will be highly dissimilar, resulting in a larger $C2I$. *In other words, when samples from different classes share similar features (reflected through their gradient alignment), we observe an increase in $C2I$.*

In what follows, we demonstrate that in the case of logistic regression, maximizing $C2I$ induces a feature averaging effect. This, in turn, generates samples that lie nearer to the decision boundary, making them more challenging to classify.

Note that we present results in terms of the alignment gap $|\mu_0 - \mu_1|$ instead of $C2I$ so that the calculations are simpler and easier to interpret[2].

**Theorem 1.** *Consider a Logistic Regression model with output probabilities $p(x) = Sigmoid(w^\top x)$, $x, w \in \mathbb{R}^d$ and cross-entropy loss $\ell(x)$. Let $\mathcal{V}_0$ and $V_1$ be validation sets containing $N/2$ samples from class 0 and class 1 respectively and $\mathcal{V} = \mathcal{V}_0 \cup \mathcal{V}_1$ (for even $N$). consider the cosine similarity between the loss gradients of a sample $x$ and a validation sample $v_i$ as:*

$$\mathcal{A}(x, v_i) = \frac{\nabla\ell(x) \cdot \nabla\ell(v_i)}{\|\nabla\ell(x)\|\,\|\nabla\ell(v_i)\|}, \tag{20}$$

*and define $\mu_c(x) = \frac{1}{N}\sum_{i \in \mathcal{V}_c} \mathcal{A}(x, v_i)$. Then the sample $x$ that maximizes the absolute class-alignment gap $|\mu_0(x) - \mu_1(x)|$ is given by the convex combination*

$$x^\star = \sum_{v_i \in \mathcal{V}} \alpha_i v_i, \qquad \alpha_i = \frac{1/\|v_i\|}{\sum_{v_j \in \mathcal{V}} 1/\|v_j\|}. \tag{21}$$

*Proof.* The mean cosine similarity to each class is:

$$\mu_0(x) = \frac{1}{N}\sum_{v_i \in D_0} \mathcal{A}(x, v_i), \qquad \mu_1(x) = \frac{1}{N}\sum_{v_i \in D_1} \mathcal{A}(x, v_i).$$

---

[2]Finding the optimal solution with $C2I$ leads to finding the roots of a cubic polynomial, which although analytically solvable, gives little insight about the nature of the solution.

Using Lemma 1, the alignment gap is:

$$|\mu_0 - \mu_1| = \left| \frac{1}{N} \sum_{v_i \in D_0} \mathcal{A}(x, v_i) - \sum_{v_i \in D_1} \mathcal{A}(x, v_i) \right| = \left| \frac{1}{N} \sum_{v_i \in D} s(v_i) \mathcal{A}(x, v_i) \right|$$

$$= \left| \frac{1}{N} \sum_{v_i \in D} s(x) s(v_i)^2 \frac{x \cdot v_i}{\|x\| \|v_i\|} \right| = \frac{1}{N} \frac{1}{\|x\|} \left| \sum_{v_i \in D} \frac{x \cdot v_i}{\|v_i\|} \right|, \qquad (22)$$

where we used that $s(x)^2 = 1$. To find the vector $x$ that maximizes the quantity above, note that the bias parameter is absorbed into $w$, so that $x$ has the form:

$$x = \begin{bmatrix} \tilde{x} \\ 1 \end{bmatrix}. \qquad (23)$$

Therefore,

$$f(\tilde{x}) \equiv (\mu_0 - \mu_1)^2 = \frac{(\tilde{x} \cdot a + \beta)^2}{\|\tilde{x}\|^2 + 1}, \qquad (24)$$

in which

$$a = \frac{1}{N} \sum_{v_i \in D} \frac{\tilde{v}_i}{\sqrt{\|\tilde{v}_i\|^2 + 1}}, \qquad \beta = \frac{1}{N} \sum_{v_i \in D} \frac{1}{\sqrt{\|\tilde{v}_i\|^2 + 1}}. \qquad (25)$$

The maximum of $f(\tilde{x})$ can be found by setting the first derivative to zero:

$$\tilde{x}^\star = \frac{a}{\beta}. \qquad (26)$$

Defining

$$\alpha_i = \frac{1}{\beta} \frac{1}{\sqrt{\|\tilde{v}_i\|^2 + 1}}, \qquad (27)$$

we have

$$\tilde{x}^\star = \sum_{v_i \in D} \alpha_i \tilde{v}_i. \qquad (28)$$

For all $v_i$, since the last dimension is equal to 1 according to Eq. equation 23, and since $\sum_i \alpha_i = 1$, it follows that the last dimension of $x^\star$ is one too. Thus, we can write the final result in terms of $x$ and $v_i$ as:

$$x^\star = \sum_{v_i \in D} \alpha_i v_i, \qquad \alpha_i = \frac{1/\|v_i\|}{\sum_{v_j \in D} 1/\|v_j\|}. \qquad (29)$$

$\square$

To interpret this result, let us consider a case where $v_i = \bar{v} + \delta v_i$ and $\max_i \|\delta v_i\| / \|\bar{v}\| = \varepsilon$ for some $0 < \varepsilon < 1$. This condition describes a situation where deviations from the mean vector are smaller than the magnitude of the mean vector. In this case,

$$\frac{1}{\|v_i\|} = \frac{1}{\|\bar{v}\|} + \frac{\delta v_i \cdot \bar{v}}{\|\bar{v}\|^2} + \mathcal{O}(\varepsilon^2). \qquad (30)$$

As a result,

$$\alpha_i = \frac{\frac{1}{\|\bar{v}\|} + \frac{\delta v_i \cdot \bar{v}}{\|\bar{v}\|^3} + \mathcal{O}(\varepsilon^2)}{\frac{N}{\|\bar{v}\|} + \frac{\sum_i \delta v_i \cdot \bar{v}}{\|\bar{v}\|^3} + \mathcal{O}(\varepsilon^2)} = \frac{1}{N} + \frac{\delta v_i \cdot \bar{v}}{\|\bar{v}\|^2} + \mathcal{O}(\varepsilon^2). \qquad (31)$$

We can approximate $x^\star$ as follows.

$$x^\star = \sum_{v_i \in D} \alpha_i v_i = \bar{v} + \sum_{v_i \in D} \frac{(\bar{v} + \delta v_i) \delta v_i \cdot \bar{v}}{\|\bar{v}\|^2} + \mathcal{O}(\varepsilon^2) = \bar{v} + \mathcal{O}(\varepsilon^2). \qquad (32)$$

Here, we used the fact that the contribution of $\bar{v}$ in the numerator cancels out when summed over $v_i$, and we are left with two factors of $\delta v_i$ which is $\mathcal{O}(\varepsilon^2)$. Therefore, the deviation of $x^\star$ from the dataset average is small. This result helps us in interpreting $x^\star$ in terms of "hard examples". In the following, we show that dataset average has a high classification loss because it is closer to the decision boundary than each cluster average.

**Lemma 2.** *Let a binary logistic regression model predict the probability of class 1 via*

$$\hat{p}(x) = Sigmoid(w^\top x),$$

*for $w \in \mathbb{R}^d$. Let $\mu_0$ and $\mu_1$ denote the means of the class-0 and class-1 inputs, respectively, and assume*

$$\hat{p}(\mu_0) = \frac{1}{2} - \varepsilon_0, \quad \hat{p}(\mu_1) = \frac{1}{2} + \varepsilon_1,$$

*for some $0 < \varepsilon_0, \varepsilon_1 < 1/2$. Let $\bar{\mu} = \pi\mu_1 + (1-\pi)\mu_0$ be the overall dataset mean for class prior $\pi \in (0, 1)$.*

*Then the model's predicted probability of the correct label at $\bar{\mu}$ is strictly less than at either class mean:*

$$\Pr(\hat{y} = 1 \mid \bar{\mu}) < \Pr(\hat{y} = 1 \mid \mu_1), \quad \Pr(\hat{y} = 0 \mid \bar{\mu}) < \Pr(\hat{y} = 0 \mid \mu_0), \tag{33}$$

*and the classification loss is lower bounded by*

$$\ell(\bar{\mu}) > \log\left(\frac{2}{1 + 2\max(\varepsilon_0, \varepsilon_1)}\right), \qquad 0 < \varepsilon_0, \varepsilon_1 < \frac{1}{2}. \tag{34}$$

*Proof.* Let $z_0 = w^\top \mu_0$, $z_1 = w^\top \mu_1$, and $\delta = w^\top \bar{\mu} = \pi z_1 + (1-\pi)z_0$, where $0 < \pi < 1$ is the ratio of class 1 number of samples to the size of the whole dataset. By assumption,

$$\hat{p}(\mu_0) = \mathrm{Sigmoid}(z_0) = \frac{1}{2} - \varepsilon_0 \quad \Rightarrow \quad z_0 < 0. \tag{35}$$

$$\hat{p}(\mu_1) = \mathrm{Sigmoid}(z_1) = \frac{1}{2} + \varepsilon_1 \quad \Rightarrow \quad z_1 > 0. \tag{36}$$

This assumption means that the model has a roughly correct guess about the class, as is the case with pretrained models. Since $\delta$ is a strict convex combination of $z_0$ and $z_1$, we have $z_0 < \delta < z_1$. By strict monotonicity of the sigmoid function, it follows that $\mathrm{Sigmoid}(z_0) < \mathrm{Sigmoid}(\delta) < \mathrm{Sigmoid}(z_1)$, i.e.,

$$\frac{1}{2} - \varepsilon_0 < \hat{p}(\bar{\mu}) < \frac{1}{2} + \varepsilon_1. \tag{37}$$

From the inequality above, we arrive at the final conclusion about model confidence

$$\Pr(\hat{y} = 1 \mid \bar{\mu}) = \hat{p}(\bar{\mu}) < \hat{p}(\mu_1) = \Pr(\hat{y} = 1 \mid \mu_1), \tag{38}$$

$$\Pr(\hat{y} = 0 \mid \bar{\mu}) = 1 - \hat{p}(\bar{\mu}) < 1 - \hat{p}(\mu_0) = \Pr(\hat{y} = 0 \mid \mu_0). \tag{39}$$

Therefore, the prediction at the mixture mean $\bar{\mu}$ is strictly less confident than at either class mean. To find the lower bound for the loss, note that if a data point at $\bar{\mu}$ has label $y = 0$, then its cross-entropy loss lower bounded by

$$\ell(\bar{\mu}; y = 0) = -\log(1 - \hat{p}(\bar{\mu})) > -\log(\frac{1}{2} + \varepsilon_0). \tag{40}$$

Otherwise, if a data point at $\bar{\mu}$ has label $y = 1$, then its cross-entropy loss lower bounded by

$$\ell(\bar{\mu}; y = 1) = -\log(\hat{p}(\bar{\mu})) > -\log(\frac{1}{2} + \varepsilon_1). \tag{41}$$

Therefore

$$\ell(\bar{\mu}) > \min\left(-\log(\frac{1}{2} + \varepsilon_0), -\log(\frac{1}{2} + \varepsilon_1)\right) = \log\left(\frac{2}{1 + 2\max(\varepsilon_0, \varepsilon_1)}\right). \tag{42}$$

$\square$

In the following, we consider extensions to more general cases.

### A.3    EXTENSION TO MULTI-CLASSIFICATION TASKS

Although we have laid out our method in a binary classification setting, its generalization to multi-class problems is straightforward. We define the Class-Contrastive Influence as the softmax over mean influence functions defined in equation 4. Specifically, if indexes $a$ and $b$ are classes that the training sample $x$ and the validation sample $v$ belong to, the average influence score is

$$\mu_{ab}(\mathbf{x}_a, \mathcal{V}_b) \triangleq \frac{1}{N} \sum_{j=1}^{N} \mathcal{A}^{\phi}(\mathbf{x}_a, v_b^j). \tag{43}$$

We define the Class-Contrastive Influence for a training sample $x_a$ belonging to class $a$ as:

$$C2I(x_a) = \text{Softmax}(\mu_{ab}) = \frac{\exp(\mu_{aa})}{\sum_b \exp(\mu_{ab})}. \tag{44}$$

This quantity will be our reward for the RL fine-tuning of the diffusion model.

This formulation reduces in the special case of binary classification to a similar formula as $D_{\text{align}}$ in equation 5. Note that in the binary case,

$$C2I(x_a) = \text{Sigmoid}(\mu_{aa} - \mu_{ab}), \tag{45}$$

which is consistent with maximizing the influence gap as proposed in the paper. Our experiments show that this reward function leads to a stable RL optimization.

### A.4    ANALYZING $x^*$ OF $C2I$ IN GENERAL FEATURE EXTRACTORS

Consider a binary classification problem with model

$$p(x) = \text{Sigmoid}(z(x; \phi)), \qquad z(x; \phi) = w^\top h(x; \phi). \tag{46}$$

Then, following similar steps as in the proof for Theorem 1, we find that

$$|\mu_0 - \mu_1| = \left| \frac{1}{N} \sum_{v_i \in D} s(v_i) \mathcal{A}(x, v_i) \right| = \frac{1}{N} \frac{1}{\|\nabla z(x; \phi)\|} \left| \nabla z(x; \phi) \cdot \sum_{v_i \in D} \frac{\nabla z(v_i; \phi)}{\|\nabla z(v_i; \phi)\|} \right|. \tag{47}$$

According to Eq. equation 47, the influence gap $|\mu_0 - \mu_1|$ is maximized if the gradient of the logit $z(x; \phi)$ is aligned with the average of $z(v_i; \phi)$ over the validation set. This result obviously reduces to the statement in Theorem 1 where in Logistic Regression we have $z = w^\top x$. For a general model, finding the input $x^\star$ with the maximal influence gap is analytically intractable, as it involves the gradient of $z$ w.r.t. to input $x$. That said, an upper bound for $|\mu_0 - \mu_1|$ will be realized if

$$\nabla z(x; \phi) \propto \sum_{v_i \in D} \frac{\nabla z(v_i; \phi)}{\|\nabla z(v_i; \phi)\|}. \tag{48}$$

## B    EXPERIMENTAL SETUP AND ADDITIONAL RESULTS

### B.1    EXPERIMENT DETAILS FOR FIGURE 3B, FIGURE 4D AND FIGURE 6.

We first generate synthetic images and create 10 distinct sets for each class, each composed of 20 images. Then, we construct a total of $10 \times 10 = 100$ training sets. Each set is combined with the few-shot original samples used to fine-tune Stable Diffusion (SD), and we train a classification model on each combination. We log both the validation loss and validation AUC at the iteration that achieves the lowest validation loss. When computing $\Gamma$ in equation 2, we use a batch size of 20, ensuring that all samples belong to the same class. In contrast, for computing $\nabla \ell$ in Eq. equation 2, we use a batch size of 1. We observe that $C2I$ is positively correlated with the validation AUC and negatively correlated with the validation loss, as shown in Figure 3b. In contrast, *influence* does not exhibit such correlations, as also shown in Figure 6. After RL fine-tuning, we generate new images using the RL-optimized SD model and repeat the procedure described above. Figure 4d demonstrates that RL fine-tuning increases $C2I$, leading to higher validation AUC and lower validation loss.

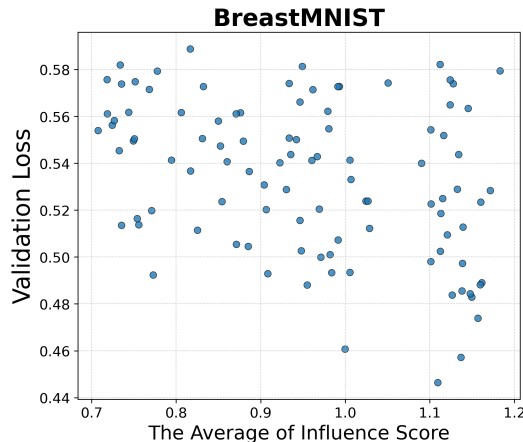

Figure 6: The average of original influence scores does not corrleate with validation loss.

## B.2 DIRECT APPLICATION OF INFLUENCE SCORE TO CLASSIFICATION TASK

We compute the average influence score for each class (equation 2) and examine its correlation with validation loss (See Appendix B for details). As shown in Figure 6, no consistent relationship is observed, suggesting influence alone is insufficient in this setting.

## B.3 MAIN EXPERIMENTS

**DermaMNIST: Multi-class and Binary Setup**  For DermaMNIST, we consider two evaluation scenarios. First, we use the original 7-class (multi-class) setting as-is. Second, we construct a binary setting by restricting evaluation to two clinically relevant classes—benign and malignant melanocytic lesions, following standard practice suggested in Ali et al. (2021); Tahir et al. (2023). This recasts DermaMNIST as a binary classification task.

**Dataset splits and statistics.**  We provide the data statistics used for validation and testing in Table 5. When training the diffusion model via RL, we use a class-balanced validation set. This is constructed by randomly sampling $\min(|\mathcal{V}_0|, |\mathcal{V}_1|)$ examples from each class. In contrast, when training the classification model, we use the full (original) validation set without balancing.

Table 5: Number of samples per class (label 0 / label 1) in the balanced validation set, original validation set, and test set.

| Dataset | Balanced validation set | Original validation set | Test set |
|---|---|---|---|
| BreastMNIST | 21 / 21 | 21 / 57 | 42 / 114 |
| PneumoniaMNIST | 135 / 135 | 135 / 389 | 234 / 390 |
| DermaMNIST-binary | 111 / 111 | 671 / 111 | 1341 / 223 |

**Results with a smaller validation set.**  For PneumoniaMNIST, the validation set is relatively large (Table 5). To examine the effect of validation size, we conducted an additional experiment by reducing it to 16 samples per class, matching the number of few-shot training samples. *As shown in Table 6, even with a much smaller validation set, our method maintains comparable performance, demonstrating robustness to validation set size.*

**Diffusion model fine-tuning.**  In RL fine-tuning, we fine-tune the diffusion model using a RL framework adapted from Black et al. (2023). We present hyperparameters used in Table 7.

**Classification model fine-tuning.**  In Pretraining, we fine-tune a ViT-B/16 model pre-trained on ImageNet using LoRA for 20 epochs on the same few-shot subset. We set the LoRA rank and

| Model | Ours (135 val/class) | Ours (16 val/class) |
|---|---|---|
| ViT | 0.945 | 0.946 |
| ResNet-50 | 0.930 | 0.937 |
| ResNet-18 | 0.956 | 0.946 |
| **Average** | 0.944 | 0.943 |

Table 6: Classifier performance on PneumoniaMNIST with different validation set sizes for RL fine-tuning.

Table 7: Hyperparameters used in diffusion model RL fine-tuning.

| Component | Value / Setting |
|---|---|
| Backbone model | Stable Diffusion 2.1 fine-tuned in Step 1 pretraining |
| LoRA rank | 16 |
| LoRA $\alpha$ | 16 |
| Mixed precision | float 16 |
| The number of inference steps | 50 |
| ETA | 0.1 |
| Guidance scale | 0.2 |
| Learning rate | $3 \times 10^{-4}$ |
| Batch size | 28 |
| Clip range | $1 \times 10^{-4}$ |

alpha to 16, and use a LoRA dropout rate of 0.1. When training with augmented data, we use the augmented training sets generated by different augmentation methods to train classification models for downstream evaluation. To assess the generalizability of our approach, we test two architectures: ViT-B/16, which was used during RL fine-tuning, and ResNet18, which was not. The ViT-B/16 model is initialized with ImageNet pretraining, whereas the ResNet18 model is trained from scratch. The hyperparameters used for each model are summarized in Table 8.

Table 8: Hyperparameters used for classifier training.

| Component | ViT-B/16 | ResNet18 |
|---|---|---|
| Initialization | LoRA fine-tuning | Trained from scratch |
| LoRA rank / $\alpha$ | 16 / 16 | – |
| Batch size | 32 | 32 |
| Learning rate | $5 \times 10^{-4}$ | $1 \times 10^{-4}$ |
| Warm-up epochs | 5 | 5 |
| LR scheduler | Linear decay | Linear decay |
| Epochs | 100 | 100 |
| Early stopping criterion | Validation AUC | Validation AUC |

For the baseline methods RandAugment and RandomErasing, we use the PyTorch transforms implementations: `RandAugment` and `RandomErasing`, respectively, with their default arguments.

**Baselines.** DataDream Kim et al. (2024) is our primary baseline which provides the starting checkpoint for RL fine-tuning. We also include Dataset Expansion Zhang et al. (2023) and DistDiff Zhu et al. (2024), diffusion-based augmentation methods that incorporates noise injection and classifier guidance; we use the original implementation. For transformation-based data augmentation, we evaluate RandAugment Cubuk et al. (2020), RandomErasing Zhong et al. (2020) and Mixup Zhang et al. (2017).

**Comparison with the simple baseline to make hard examples.** We implemented a simple baseline where Stable Diffusion is fine-tuned using only the validation images that were misclassified by the classifier. Specifically, we first trained a classifier using few-shot examples, and then identified the misclassified validation samples. For instance, in the `BreastMNIST` dataset, the validation set contains 21 samples per class. The trained classifier misclassified 15 of these samples—9 from class

0 and 6 from class 1. We used these 15 misclassified images to fine-tune Stable Diffusion and then generated 500 synthetic images per class, consistent with our main experimental setup. The results of this baseline are provided below in Table 9.

Table 9: Comparison of AUC scores on BreastMNIST between the original classifier, the simple misclassified-sample baseline, and our proposed method.

| Dataset | Method | AUC |
|---|---|---|
| BreastMNIST | Original Only | 0.828 |
| BreastMNIST | Simple Baseline (misclassified samples) | 0.778 |
| BreastMNIST | Our Method | **0.885** |

**Training sample size for BreastMNIST.** For BreastMNIST, we use 32 samples per class instead of 16 to ensure reliable classifier training. Preliminary experiments showed that using only 16 samples per class resulted in significantly lower classification performance (e.g., accuracy around 66%), compared to DermaMNIST (75%) and PneumoniaMNIST (87%). Since this classifier serves as the reward model for RL fine-tuning, lower performance on BreastMNIST led to unreliable gradient-based reward signals. Increasing the sample size to 32 per class resulted in 70% accuracy and yielded a more stable classifier for use in RL fine-tuning.

**Experiment details for Figure 5.** We first synthesize 500 samples using various data augmentation techniques, following the practical setup in Kim et al. (2024). For experiments involving fewer than 500 samples, we randomly sample 5 different subsets using different random seeds and train a separate classification model for each subset.

**Computing resources.** All reinforcement learning (RL) training experiments are conducted using a single NVIDIA A100 GPU with 40GB of memory.

**Generated samples.** Figure 7 provides representative synthetic images generated by our method, which are later used to augment BreastMNIST and DermaMNIST-binary training.

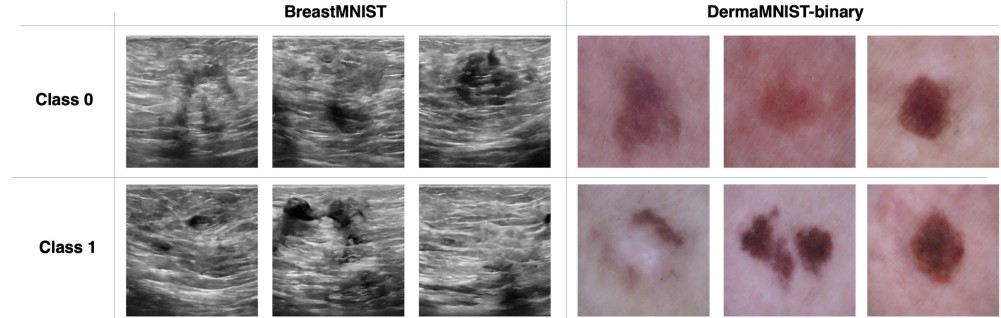

Figure 7: Examples of synthetic images generated by our method for BreastMNIST and DermaMNIST-binary. Each column shows samples from one class, illustrating that the generated images capture class-distinctive visual patterns while maintaining diversity within each class.

