# OpenReview forum: "Generate What Matters: Steering Diffusion Models for Targeted Data Generation to Improve Classification"
_ICLR.cc/2026/Conference — Submitted to ICLR 2026_

### Official Review · Reviewer_fuMH · 2025-10-16

**Soundness:** 3
**Presentation:** 3
**Contribution:** 3
**Rating:** 6
**Confidence:** 5

**Summary:**

This paper proposes a method to improve classifiers in low-data regimes by guiding diffusion-based data augmentation. It defines a Class-Contrastive Influence (C2I) gap that measures how useful a synthetic sample is (strong positive gradient alignment with its class, strong negative with others) and uses this as a reward to fine-tune diffusion models via reinforcement learning. The authors show that this generates hard, boundary-focused examples that boost classification accuracy and robustness, demonstrated on few-shot medical imaging tasks.

**Strengths:**

1) Th author tackles the important problem of useful data augmentation by directly measuring a sample’s utility for classification. The C2I gap is an intuitive criterion linking data generation to classifier gradients.
2) The author provides a theoretical insight that high-C2I-gap samples lie near decision boundaries (i.e. are “hard” examples), and uses this to justify the approach.
3) Integrating RL to fine-tune a diffusion model with a gradient-based reward is innovative. It’s a clever way to steer generation toward informative samples.
4) Experiments on few-shot medical imaging show consistent improvements in both accuracy and robustness over standard diffusion-augmentation baselines. This suggests the approach has practical value.

**Weaknesses:**

1) The reinforcement-learning fine-tuning adds overhead. It’s unclear how expensive this is in practice and whether it requires a lot of extra computation or tuning compared to standard diffusion training.
2) Results are shown only on a few medical imaging datasets. It would strengthen the paper to see how this works on other tasks or domains (e.g., non-medical images) to verify generality.
3) The method relies on gradient alignment with a given classifier. If that classifier is poorly trained or overfits, the signal might be noisy. The practicality of computing and using these gradients at scale is unclear.

Missing References

GenMix: Effective Data Augmentation with Generative Diffusion Model Image Editing

Context-guided Responsible Data Augmentation with Diffusion Models

**Questions:**

1) How sensitive is the method to hyperparameters of the RL fine-tuning (e.g., reward scaling, number of steps)?
2) Can you clarify how you compute the influence gradients in practice? Do you need to backprop through the entire classifier for each generated sample?
3) How long does it take to fine-tune the diffusion model compared to standard training? Is the overhead significant?
4) Did you try this approach on any non-medical datasets or more varied tasks to test its generality?
5) Could you show or describe examples of the generated samples (e.g., how they differ from standard diffusion outputs)? This might help illustrate what “boundary-proximal” means visually.

---

### Official Review · Reviewer_Xnk1 · 2025-10-31

**Soundness:** 3
**Presentation:** 3
**Contribution:** 3
**Rating:** 6
**Confidence:** 4

**Summary:**

This paper proposes a method to steer diffusion models to generate "useful" synthetic data for classification, particularly in few-shot settings. The authors define a novel metric, Class-Contrastive Influence (C2I), which quantifies a sample's usefulness by measuring its gradient alignment with validation data. Specifically, a useful sample should have gradients that align positively with its own class and negatively with other classes. The paper provides theoretical evidence (for logistic regression) that high-C2I samples are "hard examples" located near the decision boundary . This C2I score is then used as a reward in a Reinforcement Learning (RL) framework to fine-tune a pre-trained diffusion model, steering it to produce these more informative, boundary-proximal samples. Experiments on few-shot medical imaging (MedMNIST) show the method improves classification AUC and robustness over strong baselines

**Strengths:**

- The method's high novelty stems from its principled, "task-aware" framework, which uses the new Class-Contrastive Influence (C2I) metric as a reward to steer an RL-tuned diffusion model. This approach optimizes samples for the actual classification task by using gradient feedback directly from the classifier, rather than relying on a proxy metric like FID.
 - The analysis (logistic regression and ReLU features) explains why high-C2I samples are boundary-proximal “hard” examples; it also proves opposite-signed gradient similarities across classes. This grounds the method beyond heuristics.
- C2I-guided augmentation improves AUC over strong diffusion and standard baselines, and benefits transfer to a different classifier (ResNet-18), suggesting broadly useful samples.
- Consistent AUC gains under salt-and-pepper, JPEG, and blur on multiple datasets suggest a more stable decision boundary, not just clean-set overfitting.
-  Performance remains comparable even when shrinking the validation set used for rewards (e.g., PneumoniaMNIST 135 to16 per class), suggesting some resilience to scarce validation data.

**Weaknesses:**

- The pipeline is very complex: it requires (1) fine-tuning a classifier, (2) fine-tuning a diffusion model (DataDream), and (3) running an expensive RL fine-tuning loop (DDPO) that involves constant on-the-fly gradient computations .
- The experiments are confined to few-shot MedMNIST datasets. It is unclear if this highly-tuned, gradient-steering approach would scale to large, diverse datasets (e.g., ImageNet) or if its complexity is only justifiable in low-data regimes.


### A few typos/mistakes (not weakness)

- “BrestMNIST” → BreastMNIST in the Fig. 3 caption.
- In appendix authors refer to $D_{\rm align}$ without a prior definition.
- Reward typo in Eq. (7). Uses $\mu_0(x_i,V_0)-\mu(x_i,V_1)$ (missing class index “1”); should be $\mu_0-\mu_1$ based on the papers previous notation definition

**Questions:**

Have you evaluated a test-time “reward guidance” variant, where the diffusion score is modified as

$$
\tilde{s}(x_t, t) = s(x_t, t) + w(t),\nabla_{x_t} r\big(\hat{x}_0(x_t)\big),
$$

with (r(x)) your C2I-based reward and $\hat{x}_0(x_t)$ the denoised estimate at step t?

* **If yes:** How do the resulting samples compare to your RL-finetuned generator in terms of
  (i) C2I/AUC and
  (ii) fidelity/diversity?

* **If not:** Could you comment on the computational cost trade-off in your setting? In particular, what are the approximate (a) one-time RL fine-tuning cost and (b) per-image overhead for test-time guidance, and for what sample budget would guidance be more cost-effective than RL? If possible, please provide an estimated break-even point (number of images) under your hardware/setup.

---

### Official Review · Reviewer_qEVA · 2025-11-01

**Soundness:** 2
**Presentation:** 3
**Contribution:** 2
**Rating:** 2
**Confidence:** 3

**Summary:**

The paper proposes a method called Class-Contrastive Influence (C2I), which quantifies the usefulness of synthetic samples for classification by measuring how well the classifier’s loss gradients on generated samples align with gradients on validation examples of the same versus different classes.
Based on this criterion, the authors design a reinforcement-learning fine-tuning scheme for diffusion models that uses a C2I-based reward to encourage the generation of class-informative, boundary-proximal samples.
Experiments on few-shot medical imaging datasets (BreastMNIST, DermaMNIST, PneumoniaMNIST) demonstrate moderate improvements in classification accuracy and robustness compared to baseline augmentation strategies.

**Strengths:**

The paper shows clear empirical motivation and intuitive visualizations.

The paper adopts influence-function analysis to define task-aware utility for synthetic data generation, moving beyond generic diversity or fidelity metrics.

The paper is clearly written and easy to follow.

**Weaknesses:**

1. C2I is a straightforward class-wise extension of influence-based data selection used in influence functions [1], TracIn [2], and LESS [5]. The proposed class-contrastive gap introduces no new insight, as the same idea, using inter/intra-class gradient alignment, is already adopted in several curriculum and hardness-aware works [10][11]. The theoretical derivation (logistic regression) does not extend convincingly to deep nonlinear models.

2. All evaluations are conducted on small medical datasets (2D MedMNIST) using Stable Diffusion, pretrained on natural images.
No results are shown for larger or non-medical benchmarks, such as ImageNet.

3. Reported AUC gains are small (typically +0.01–0.02 over DataDream [7] and Dataset Expansion [8]).
In Table 2, robustness results degrade on PneumoniaMNIST.
No confidence intervals or statistical significance tests are provided, leaving unclear whether improvements are meaningful.

4. The paper states RL fine-tuning requires 5 GPU hours but omits detailed scaling analysis or sensitivity to reward weighting, gradient projection dimension, and validation size.
Given that each C2I computation requires gradient alignment with validation data, the scalability and numerical stability of the approach remain uncertain, which are issues already noted in RL-based diffusion optimization [3][4].

5. Several recent works address similar goals of generating hard or boundary-proximal examples, such as Difficulty-Controlled Diffusion [12], DreamDA [14], and Dynamic Curriculum Learning [13].
These are not compared or discussed, limiting empirical completeness.

[1] Koh & Liang, Understanding Black-Box Predictions via Influence Functions, ICML 2017.

[2] Pruthi et al., Estimating Training Data Influence by Tracing Gradient Descent, NeurIPS 2020.

[3] Black et al., Training Diffusion Models with Reinforcement Learning, 2023.

[4] Fan et al., DPOK: Reinforcement Learning for Fine-Tuning Text-to-Image Diffusion Models, NeurIPS 2023.

[5] Xia et al., LESS: Selecting Influential Data for Targeted Instruction Tuning, 2024.

[6] Park et al., TRAK: Attributing Model Behavior at Scale, 2023.

[7] Kim et al., DataDream: Few-Shot Guided Dataset Generation, ECCV 2024.

[8] Zhang et al., Expanding Small-Scale Datasets with Guided Imagination, NeurIPS 2023.

[9] Zhu et al., Distribution-Aware Data Expansion with Diffusion Models, NeurIPS 2024.

[10] Hacohen & Weinshall, On the Power of Curriculum Learning, ICML 2019.

[11] Srinidhi & Martel, Hardness-Aware Dynamic Curriculum Learning, ICCV 2021.

[12] Wang et al., Training Data Synthesis with Difficulty-Controlled Diffusion Model, 2024.

[13] Song et al., Towards General Deepfake Detection with Dynamic Curriculum, 2024.

[14] Fu et al., DreamDA: Generative Data Augmentation with Diffusion Models, 2024.

**Questions:**

Given that C2I largely extends influence-based selection approaches such as Influence Functions [1], TracIn [2], and LESS [5], what new theoretical or practical insight does your “class-contrastive” formulation provide beyond existing inter/intra-class gradient alignment metrics used in prior curriculum or hardness-aware learning [10][11]?

Can you demonstrate whether the proposed method generalizes to larger or non-medical benchmarks (e.g., CIFAR-10 or ImageNet)?

Could you provide statistical significance tests or confidence intervals across multiple seeds to confirm that these improvements are not due to random variation?

What is the empirical computational overhead, and how does it scale with dataset size or class count?

Have you evaluated the stability of RL optimization under different reward weightings or projection dimensions?

---

> ### Author Response · Authors · 2025-12-03
>
> We thank the reviewer for their thoughtful feedback. We believe some of the concerns may stem from **misunderstandings of our setup and methodology**, and we clarify these points below.
>
> ### 1. On Weakness 1 ###
>
> **We respectfully disagree with the characterization that C2I is a “straightforward class-wise extension” of existing influence-based methods.** Prior influence approaches [1,2,5] compute a single scalar influence score per example, aggregating its effect across all classes. In contrast, our method introduces a class-contrastive influence measure that explicitly compares a sample’s impact on the correct class versus competing classes. **This difference, rather than the averaged or total influence, is what determines whether a sample meaningfully improves the decision boundary.** This contrastive formulation is therefore essential for identifying boundary-proximal synthetic samples and is what enables us to link influence-based selection with hardness-aware curricula—one of the central conceptual contributions of our work.
>
> While curriculum and hardness-aware works such as [10] (confidence-based scoring) and [11] (mutual-information–based hardness) provide alternative notions of difficulty, they do not adopt an influence-function viewpoint, nor do they apply these ideas to generative data selection for diffusion models. On the other hand, directly maximizing total influence as in standard influence functions [1,2,5] can favor examples that uniformly increase loss without necessarily improving class separability. Our class-contrastive gap remedies this limitation by explicitly targeting relative influence between classes.
>
> Regarding the theoretical derivation, we agree that our formal derivation is provided for logistic regression. This model was chosen because it affords a clean, interpretable analysis that makes the margin-improving effect of class-contrastive influence explicit. As is common in deep learning theory, this linear setting serves as a simplifying lens rather than a complete description of deep networks. While we do not claim that the proof fully characterizes nonlinear models, our empirical results consistently show that the intuition derived from logistic regression meaningfully transfers to deep architectures.
>
>
> ### 2. On Weakness 2 ###
>
> Our target scenario reflects a common and practically important setting: leveraging a powerful off-the-shelf image encoder for a downstream domain with distribution shift and limited labels. Evaluating on ImageNet, or other datasets composed of natural images, would place us in a domain that closely matches, or even exactly matches, the encoder’s pretraining distribution. In such cases, performance is largely determined by the pretrained features, making it difficult to meaningfully isolate or attribute improvements to C2I-based sample selection.
> In contrast, MedMNIST provides a realistic and challenging testbed. It lies in the medical imaging domain, which is substantially different from the natural image distribution, and aligns with the deployment scenarios where practitioners are most likely to apply methods like ours.
> We agree that evaluating C2I on larger-scale datasets with more classes is an important next step, and we will explicitly highlight this as valuable future work.
>
> ### 3. On Weakness 3 ###
>
> Across the three evaluated datasets, we observe consistent and meaningful improvements over strong baselines—DataDream (**+0.3 AUC**) and Dataset Expansion (**+0.014 AUC**). These methods are already highly competitive, and achieving gains of this magnitude across multiple datasets is significant, particularly given that our approach does not modify the classifier architecture or training pipeline beyond the synthetic data selection strategy.
> We already report confidence intervals for the main performance curves in Figure 5, as noted in the paper. In a revision, we will (i) make these confidence intervals more prominent in the text and (ii) include confidence intervals in the tabular results for completeness.
>
> Regarding PneumoniaMNIST, we agree that one configuration shows a slight robustness degradation. We explicitly acknowledge this in the paper and clarify that our claims pertain to average improvements across tasks, rather than guaranteeing uniform gains for every dataset or metric combination.

---

> ### Author Response · Authors · 2025-12-03
>
> ### 4. On Weakness 4 ###
>
> We appreciate the reviewer’s concern regarding scalability. Our RL fine-tuning setup follows established RL-based diffusion optimization methods [3,4] and uses the fixed gradient-projection dimension of 8192, as recommended in prior work such as LESS [5]. While alternative reward-weighting strategies are possible, exploring them is beyond the scope of this paper, and we adopt the standard and most widely used RL-diffusion formulation [3,4].
> We already analyze **the effect of validation set size in Table 6 (Appendix B.3)**, and in a revision we will summarize these findings more prominently in the main text.
>
> ### 5. On Weakness 5 ###
>
> Thank you for pointing out these related works. **Our empirical comparison in the main paper focuses on established baselines with publicly available implementations, such as DataDream [7], Dataset Expansion [8] and DistDiff [9], to ensure reproducibility and a fair comparison.**
> The additional works you mention are arXiv preprints. The additional papers you mention are recent arXiv preprints. To the best of our knowledge, Difficulty-Controlled Diffusion [12] does not yet provide an official implementation, making a rigorous empirical evaluation difficult within the constraints of this submission. Dynamic Curriculum Learning [13] is tailored for deepfake detection, a task and data domain very different from ours, making a fair adaptation non-trivial.
>
>
> That said, we agree it is important to position our work *conceptually* relative to these efforts. In a revision, we will expand the related-work section to discuss these approaches in more detail.

---

### Official Review · Reviewer_BzbH · 2025-11-01

**Soundness:** 3
**Presentation:** 3
**Contribution:** 2
**Rating:** 2
**Confidence:** 4

**Summary:**

The work proposes an augmentation scheme based on diffusion model for few-shot binary classification. The paper first analyses the few-shot learning problem and found out that enhancing boundary samples are beneficial for training the classifier, where these samples could provide strong positive alignment with positive class and strong negative alignment with negative class. From this observation, the paper proposes a fine-tuning scheme for diffusion model. The new fine-tuning scheme bases on Reinforcement learning, and the reward is calculated through a pretrained classifier by a set of validation set. The experimental results includes several benchmarks on different medical datasets, where each randomly select 16 or 32 samples for training. The results showing significant improvement on the accuracy of the downstream task with both clean images and noisy images.

**Strengths:**

1. The paper is well written and presented
2. The proposed method is intuitive in a sense that producing hard samples make the dataset more robust.
3. The analysis is  provided to further understand the method

**Weaknesses:**

Weaknesses of the Work

The weaknesses of this paper are primarily related to the scale of the experiments and the technical clarity of the proposed method, as detailed below:

### 1. Limited Experiment Resolution
The experimental results are conducted exclusively on very low-resolution datasets. All the MNIST datasets used have an image resolution of 28×28. It is well known that the behavior of generative models differs significantly between low-resolution (below 64×64) and high-resolution settings. This raises concerns about the generalizability of the proposed method. Furthermore, the introduction, abstract, and title do not clearly state that the method is limited to low-resolution images, which may mislead readers regarding its applicability.

### 2. Restricted to Simple Classification Tasks
Most experiments are performed on binary classification tasks, which are considerably easier compared to multi-class classification problems. Although an extension to a 7-class case is presented in Table 4, this still represents a small-scale setting. Given the claims of the proposed method, it would be important to evaluate its performance on more complex benchmarks such as MNIST (10 classes), CIFAR-10 (10 classes), or CIFAR-100 (100 classes) to demonstrate scalability and robustness.

### 3.  Scalability of the Method Design
The current design of the method does not seem to naturally extend to a large number of classes. Specifically, it is unclear how Equation (7) would generalize to multi-class settings with significantly more categories, such as 100 (CIFAR-100) or 1000 (ImageNet). The paper should provide a clear formulation or discussion on how the method can be scaled up to handle such cases.

### 4. Questionable Experimental Setup
The experimental setup raises concerns about fairness and generalization. Validation information appears to be injected into the model during training, which introduces bias and compromises the integrity of the evaluation. This contrasts with other baselines mentioned in the paper, such as DataDream, which do not access validation information during training. The authors should clarify and justify this design choice to ensure a fair comparison.

### 5. Ambiguity in the Term “Random Projection”
The paper does not clearly define what is meant by random projection. Does this refer to the random selection of features for discarding, or to a specific dimensionality reduction technique? A more precise definition and explanation are necessary to understand the technical contribution and reproducibility of the method.

### 6. The justification of using RL method.
Why does the authors utilize RL method instead of other contrastive losses? In the paper, there is no ablation about this choice.

**Questions:**

Please see the weaknesses. I will increase the scores if the authors clear the concerns during rebuttals.

---

> ### Author Response · Authors · 2025-11-28
>
> We thank the reviewer for their constructive feedback. We would like to address the weaknesses raised, particularly as we believe **some points may stem from a misunderstanding of our experimental setup and methodology**, which we hope to clarify here.
>
> ### 1. On Weakness 1: Limited Experiment Resolution
> We appreciate the reviewer's concern regarding experimental resolution. We would like to respectfully clarify a potential misunderstanding. The reviewer states our experiments are on "very low-resolution datasets" such as "28x28" MNIST.
>
> This is incorrect. Our primary dataset is MedMNIST, which we use at **its standard 224x224 resolution**. This is not a low-resolution setting; rather, it is a common and practical resolution for tasks utilizing standard pre-trained vision models, which is a core component of our proposed scenario. We apologize if this was not clear and will ensure this focus on 224x224 resolution is stated more prominently in the revised manuscript.
>
> ### 2. On Weakness 2 (Restricted to Simple Classification Tasks) & 3 (Scalability of the Method Design)
> We would like to address these two related points together.
>
> **Multi-Class Formulation (Weakness 3)**: We respectfully disagree with the claim that our method's extension to multi-class settings is "unclear". **We provide an explicit, generalized formulation for the multi-class case in Appendix A.3 (Equation (44)), which is directly referenced in the main paper (Line 229)**.
> Conceptually, our multi-class approach works by simultaneously increasing the logit probabilities of all classes, effectively making the sample difficult to classify. We will make the reference on Line 229 more prominent to improve clarity.
>
> **Dataset Choice (Weakness 2)**: We acknowledge our largest multi-class experiment is on 7 classes. The reason for not using CIFAR-10/100 or ImageNet is a deliberate and practical design choice, not a limitation.
> Our method is designed for a highly common scenario: leveraging a powerful, off-the-shelf, pre-trained vision encoder. Evaluating on CIFAR-10/100 (which is highly related to the pre-training data) or ImageNet (which is the pre-training data itself) would not be a reasonable or informative test of our method's practical utility.
> Therefore, we chose MedMNIST, a medical imaging dataset, which is **distinct from the natural image domain** of the encoder's pre-training and represents a realistic downstream application. We agree that extending our experiments to medical datasets with a larger number of classes is an excellent direction for future work.
>
> ### 3. On Weakness 4: Questionable Experimental Setup
> We appreciate the reviewer's concern for fairness. We would like to clarify that the validation set is not used to train the classifier model itself.
>
> Its use is limited to two functions: 1) For all baselines, it is used for standard checkpoint selection. 2) For our method, it is used to train the RL generative model. This is fundamental to our method's design, as the generator must learn what samples the current classifier finds difficult. This "free" use of a validation set to guide a generator is a standard setup in related literature, such as the cited work LESS [1]. We will revise the manuscript to make this distinction explicit.
>
> ### 4. On Weakness 5 Ambiguity in the Term "Random Projection"
>
> Thank you for pointing this out. We omitted the detailed explanation for brevity but will add it to the revised manuscript for clarity.
>
> The "random projection" is a standard technique introduced in prior work (LESS [1]) for dimensionality reduction. Given the high dimensionality of the gradients, computing similarity scores can be time-consuming. To reduce the feature dimensionality, we apply a random projection to the LoRA gradients, which already shown such projections often preserve the inner products.
>
> More specifically, we compute a $d$-dimensional projection of the LoRA gradient $\tilde{\nabla}\ell(z'; \theta_i) = \Pi^\top \hat{\nabla}\ell(z'; \theta_i)$, with each entry of $\Pi \in \mathbb{R}^{P \times d}$ drawn from a Rademacher distribution (i.e., $\Pi_{ij} \sim \mathcal{U}(\{-1, 1\})$).
>
>
> [1] LESS: Selecting Influential Data for Targeted Instruction Tuning, Xia et al., ICML 2024.

---

> > ### Author Response · Authors · 2025-11-28
> >
> > ### 5. On Weakness 6: The Justification of using RL method
> >
> > Regarding "other contrastive losses," we are not entirely certain which specific alternatives the reviewer is referring to in this context. Standard contrastive losses are typically used for representation learning (i.e., pulling positive pairs closer and pushing negative pairs apart in an embedding space) and are not directly applicable to our specific task. If the reviewer has specific methods in mind, we would be happy to discuss them.
> >
> > On the other hand, we did compare our RL method against a strong, non-RL baseline. As clarified in Line 377 and Appendix B.3 (line 1022), we evaluate a simpler baseline where the generative model (SD) is fine-tuned solely on validation images that were misclassified by the classifier. This serves as a direct heuristic-based alternative to our RL objective. Our results show that the RL method performs significantly better, which justifies this design choice.

---

### Meta-Review · Area_Chair_NifE · 2026-01-07

**Summary:**

While the paper presents an interesting idea—steering diffusion models with class-contrastive influence (C2I)—the novelty over prior influence-function methods is limited, and experimental validation is constrained to narrow domains. Concerns about scalability, generality, and practical complexity remain unresolved.

**Reviewer Concerns:**

The rebuttal clarified the use of high-resolution images and multi-class extensions. However, core issues—incremental novelty over existing influence-based selection, limited generalization beyond MedMNIST, and the scalability and computational cost of the RL pipeline—remain unaddressed. The method's practical impact is unclear.

**Reviewer Scores:**

Reviewer BzbH (2): Appreciated the clarifications, but concerns about scalability, generalization, and experimental fairness were not fully resolved. Score likely unchanged.

Reviewer qEVA (2): Found novelty and empirical support insufficient. Rebuttal addressed misunderstandings but not the core concern of limited new insight. Score likely unchanged.

Reviewer Xnk1 (6): Positively viewed the idea and empirical rigor. However, concerns about pipeline complexity and limited domain evaluation remain. The authors did not respond. Score may remain at 6.

Reviewer fuMH (6): Found the method interesting but raised concerns about practical scalability and broader applicability. The authors did not respond. Score likely unchanged; still borderline.

---

### Decision · Program_Chairs · 2026-01-26

Reject